# Late-Evening Snack with Branched-Chain Amino Acid-Enriched Nutrients Does Not Always Inhibit Overt Diabetes in Patients with Cirrhosis: A Pilot Study

**DOI:** 10.3390/nu11092140

**Published:** 2019-09-07

**Authors:** Keisuke Nakanishi, Tadashi Namisaki, Tsuyoshi Mashitani, Kosuke Kaji, Kuniaki Ozaki, Soichiro Saikawa, Shinya Sato, Takashi Inoue, Yasuhiko Sawada, Kou Kitagawa, Hiroaki Takaya, Naotaka Shimozato, Hideto Kawaratani, Kei Moriya, Takemi Akahane, Akira Mitoro, Hitoshi Yoshiji

**Affiliations:** 1Third Department of Internal Medicine, Nara Medical University, 840 Shijo-cho, Kashihara, Nara 634-8522, Japan; nakanishi@naramed-u.ac.jp (K.N.); mashiyan66@yahoo.co.jp (T.M.); kajik@naramed-u.ac.jp (K.K.); ozaki@naramed-u.ac.jp (K.O.); saikawa@naramed-u.ac.jp (S.S.); shinyasato@naramed-u.ac.jp (S.S.); yasuhiko@naramed-u.ac.jp (Y.S.); kitagawa@naramed-u.ac.jp (K.K.); htky@naramed-u.ac.jp (H.T.); shimozato@naramed-u.ac.jp (N.S.); kawara@naramed-u.ac.jp (H.K.); moriyak@naramed-u.ac.jp (K.M.); stakemi@naramed-u.ac.jp (T.A.); mitoroak@naramed-u.ac.jp (A.M.); yoshijih@naramed-u.ac.jp (H.Y.); 2Institute for Clinical and Translational Science, Nara Medical University Hospital, 840 Shijo-cho, Kashihara, Nara 634-8522, Japan; tk-inoue@naramed-u.ac.jp

**Keywords:** liver cirrhosis, branched-chain amino acid-enriched nutrients, impaired glucose metabolism, insulin resistance, protein–energy malnutrition, a late-evening snack, type 2 diabetes mellitus

## Abstract

Cirrhosis patients often have abnormal glucose metabolism. We investigated the effects of a late-evening snack (LES) with branched-chain amino acid-enriched nutrients (BCAA-EN) on glucose metabolism in cirrhosis patients. LES with BCAA-EN was administered for 1 week in 13 patients with cirrhosis and hypoalbuminemia. Blood glucose (BG) levels were measured every 15 min. The patients were divided into two groups based on BG levels: group 1 (G1, *n* = 11): nocturnal BG levels <200 mg/dL and group 2 (G2, *n* = 2): nocturnal BG levels ≥200 mg/dL. G1 had nocturnal BG levels <200 mg/dL, whereas G2 had nocturnal BG levels ≥200 mg/dL. The average BG levels did not significantly change after BCAA-EN administration in G1 (before 91.9 ± 29.0 mg/dL; after 89.0 ± 24.3 mg/dl). However, the average BG levels significantly increased after BCAA-EN administration in G2 (before 153.6 ± 43.3 mg/dL; after 200.9 ± 59.7 mg/dL) (*p* < 0.01). The glycated albumin level (16.6 ± 0.9% vs. 16.2 ± 2.1%), fasting immunoreactive insulin (F-IRI) level (53.9 ± 34.0 μU/mL vs. 16.5 ± 11.0 μU/mL), and homeostasis model assessment of insulin resistance (HOMA-IR) score (17.85 ± 10.58 vs. 4.02 ± 2.59) were significantly higher in G2 than in G1 (*p* < 0.05, *p* < 0.05, and *p* < 0.01, respectively). The quantitative insulin sensitivity check indices (0.32 ± 0.03 vs. 0.27 ± 0.02) were significantly higher in G1 than in G2 (*p* < 0.01). One patient in G2 was obese and had type 2 diabetes. The other patient was obese and had a high HOMA-IR score and F-IRI level. A LES with BCAA-EN does not inhibit overt diabetes in most cirrhosis patients. However, close attention should be paid to fluctuations in BG levels in cirrhosis patients who present with obesity and severe insulin resistance.

## 1. Introduction

The liver is an essential organ in regulating the biological processes of glucose, protein, and energy metabolism. Protein–energy malnutrition (PEM) is the most common characteristic of and can result in various complications associated with hypoalbuminemia in patients with cirrhosis [1,2,3,4,5]. Branched-chain amino acids (BCAAs) play primary roles in albumin synthesis and energy metabolism through the phosphoinositide 3-kinase/serine-threonine protein kinase-mammalian target of the rapamycin signaling pathway [1]. Patients with cirrhosis have lower serum BCAA levels [2,3]. Two different BCAA formulations (BCAA granules [BCAA-GR: LIVACT granules, Eisai Co., Ltd., Tokyo, Japan], which contain 4 g of BCAA and 16 kcal of energy per pack [4,5], and BCAA-enriched nutrients [BCAA-EN: Aminoleban EN, Otsuka Pharmaceutical, Tokyo, Japan], which contain 5.6 g of BCAA and 210 kcal of energy per pack) were approved for the treatment of patients with cirrhosis who present with hypoalbuminemia. BCAA supplementation prevents various complications, including hepatic encephalopathy, and improves event-free survival and quality of life in patients with cirrhosis [2]. Late-evening snacks (LES) with BCAA can provide extra protein and calories at bedtime and prevent fat oxidation and catabolic state after overnight fasting [6]. Results of randomized controlled trials have revealed that nocturnal administration of BCAA-EN improves liver function and nutritional status in patients with cirrhosis [7]. Furthermore, LES is recommended for patients with cirrhosis to prevent starvation status in the morning according to the European Society for Clinical Nutrition and Metabolism [8] and the American Society for Parenteral and Enteral Nutrition guidelines [9]. However, patients with cirrhosis often have abnormal glucose metabolism. The effect of BCAA-EN administration on glucose metabolism in patients with cirrhosis has not been fully elucidated. Therefore, this study aimed to investigate the effects of LES with BCAA-EN on glucose metabolism in patients with cirrhosis.

## 2. Materials and Methods

### 2.1. Patients

The study was conducted on 160 patients with cirrhosis who presented with hypoalbuminemia between June 2015 and June 2016 at the Third Department of Internal Medicine, Nara Medical University. The inclusion criteria were as follows: (i) patients aged 20–80 years and (ii) those with compensated cirrhosis without any symptoms, such as jaundice, ascites, or hepatic encephalopathy. A total of 147 patients were excluded due to the following reasons: (i) treatment of cirrhosis with BCAA-GR (*n* = 27), (ii) presence of a viable hepatocellular carcinoma (*n* = 27), (iii) unavailability of the questionnaire (*n* = 16), (iv) death (*n* = 25) or patient censoring (*n* = 7), (v) organ dysfunction, such as kidney failure, heart failure, and lung failure, (*n* = 9) (vi) serum creatinine values ≥1.5 mg/dL) (*n* = 10), (vii) use of prescription medication that interferes with prothrombin time (e.g., warfarin) (*n* = 25), (viii) persistent alcohol consumption, and (viiii) poor palatability of BCAA-EN (*n* = 1). Blood glucose (BG) levels were determined every 15 min for 14 days (day 1–day 14) with the Flash Glucose Monitoring (FGM) system (Abbott Diabetes Care, Alameda, CA) [10] (Figure 1A). BCAA-EN was administered at bedtime for 1 week (day 8–day 14) in 13 (8.1%) of 160 patients (Figure 1B). A nocturnal blood glucose concentration of ≥200 mg/dL is the appropriate threshold for the diagnosis of diabetes [11]. The patients were divided into two groups based on BG levels: group 1 (G1, *n* = 11; nocturnal BG levels of <200 mg/dL) and group 2 (G2, *n* = 2, nocturnal BG levels of ≥200 mg/dL). Changes in BG levels before (every 15 min in the daytime and nighttime between days 1 and 7) and after (every 15 min in the daytime and nighttime between days 8 and 14) BCAA-EN administration were compared between G1 and G2. Primary outcome is suppression of elevation of BG levels after BCAA-EN administration. The estimated effect size is the differences in BG levels before and after BCAA-EN administration. The participants adhered to the BCAA-EN therapy, and their food intake did not change during the study period. The study was conducted in accordance with the Declaration of Helsinki, and a written informed consent was obtained from all study participants. The study protocol was approved by the medical ethics committee of Nara Medical University (13ken034).

### 2.2. Statistical Analysis

Data are presented as mean ± standard deviation. All statistical analyses were performed using IBM Statistical Package for the Social Sciences software version 22. The Mann–Whitney U test was used in this study for comparing the baseline characteristics between G1 (patients with nocturnal BG levels of <200 mg/dL) and G2 (patients with nocturnal BG levels of ≥200 mg/dL). The Wilcoxon signed-rank test was used to compare the changes in BG levels before and after BCAA-EN administration between the two groups.

## 3. Results

### 3.1. Clinical Characteristics of the Participants

The baseline characteristics of the participants are summarized in Table 1. Our study population comprised 7 men and 6 women with a mean age of 69.8 ± 12.3 years. Nine (69.2%) patients presented with Child–Pugh class A cirrhosis and 4 (30.8%) with class B cirrhosis. The average body mass index (BMI) of all patients was 23.7 ± 4.0 kg/m^2^. The mean serum total protein (TP), albumin (ALB), glycated albumin (GA), fasting BG (FBG), and fasting immunoreactive insulin (F-IRI) level, homeostatic model assessment insulin resistance (HOMA-IR) score, and quantitative insulin sensitivity check index (QUICK-I) score were 7.3 ± 0.5 g/dL, 3.9 ± 0.5 g/dL, 16.2% ± 2.0%, 104.9 ± 23.7 mg/dL, and 22.2 ± 21.5 μU/mL, 6.15 ± 6.91, and 0.31 ± 0.03, respectively.

G1 had nocturnal BG levels (from 12:00 a.m. to 8 a.m.) <200 mg/dL, whereas G2 had nocturnal BG levels ≥200 mg/dL. G2 was significantly younger than G1 (61.5 ± 11.5 vs. 71.2 ± 11.9 years) (*p* < 0.05). The Child–Pugh score of G2 was significantly lower than that of G1 (5.5 ± 0.5 vs. 6.1 ± 1.2) (*p* < 0.05). The TP (7.7 ± 0.1g/dL vs. 7.2 ± 0.5g/dL), ALB (4.1 ± 0.1g/dL vs. 3.9 ± 0.5g/dL), and GA levels (16.6 ± 0.9% vs. 16.2 ± 2.1%), F-IRI score (53.9 ± 34.0 μU/mL vs. 16.5 ± 11.0 μU/mL), and HOMA-IR score (17.85 ± 10.58 vs. 4.02 ± 2.59) were significantly higher in G2 than in G1 (*p* < 0.05, *p* < 0.05, *p* < 0.05, *p* < 0.05, and *p* < 0.01, respectively). The QUICK-I values (0.27 ± 0.02 vs. 0.32 ± 0.03) were significantly lower in G2 than in G1 (*p* < 0.01).

### 3.2. Clinical Profiles of the Participants

Three (23.1%) patients were obese (BMI ≥25 kg/m^2^). Nine (69.2%) patients had fasting hyperinsulinemia (serum F-IRI level ≥15 µU/mL)] [12] and 8 (61.5%) were insulin resistant (HOMA-IR score ≥2.5) [13]. All patients had QUICK-I values <0.38; this value is considered the cutoff value for insulin sensitivity [14]. As in patients 9–11 in Table 2, 3 (27.3%) of 11 patients who were diagnosed with type 2 diabetes were treated with antidiabetic agents. Patient 12 was obese and had type 2 diabetes (hemoglobin A1c: 7.2% and FBG level: 148 mg/dL). Patient 13 was obese and had a high HOMA-IR score at 28.2 and an elevated F-IRI level at 87.9 µU/mL. However, none of the patients had hypoglycemia during early morning. These findings indicated that the clinical characteristics of G2 whose nocturnal BG levels were <200 mg/dL include obesity and severe insulin resistance.

### 3.3. The Changes in BG Levels before and after Branched-Chain Amino Acid Enteral Nutrient Administration

The average BG levels did not significantly change after BCAA-EN administration in G1 (before 91.9 ± 29.0 mg/dL; after 89.0 ± 24.3 mg/dL) (Figure 2A). However, the average BG levels significantly increased after BCAA-EN administration in G2 (before 153.6 ± 43.3 mg/dL; after 200.9 ± 59.7 mg/dL) (*p* < 0.01) (Figure 2B).

## 4. Discussion

LES with BCAA has been recommended as a nutritional treatment for patients with cirrhosis who present with hypoalbuminemia [15]. However, metabolic abnormalities, including PEM and deteriorated glucose metabolism, were observed with the high prevalence of liver cirrhosis. To the best of our knowledge, this study was the first to show that LES with BCAA-EN does not always inhibit overt diabetes in patients with cirrhosis. A significant increase in the average BG levels was found in patients with nocturnal BG levels ≥200 mg/dL after LES treatment. The results of the present study indicated that obesity and severe insulin resistance are the possible risk factors for the deterioration of glucose homeostasis in patients with cirrhosis treated with LES with BCAA-EN. Several studies have shown that the administration of LES for 1 week improves energy malnutrition and glucose tolerance in patients with cirrhosis [16,17,18]. Supplementation with BCAA-EN improves insulin resistance in patients with decompensated cirrhosis [19]. In contrast, recent evidence has shown that BCAA supplementation contributes to the development of insulin resistance [20] and increases the risk of incident type 2 diabetes [21]. BCAA-EN administration increases glucose metabolism in patients with cirrhosis who have impaired glucose tolerance [22]. The discrepancy between the results of the present and previous studies might be attributed to the differences in the duration of BCAA administration and the liver functional reserve of the participants. The present study included patients with compensated cirrhosis without any symptoms, such as jaundice, ascites, or hepatic encephalopathy, whereas another clinical study included patients with cirrhosis and worse liver function [22]. Nishikawa et al. have shown that the proportion of patients with PEM increases as the hepatic functional reserve of patients with cirrhosis worsens [23]. BCAA supplementation is effective in improving protein malnutrition in patients with cirrhosis, irrespective of liver disease severity [2]. These findings may explain why the Child–Pugh score and ALB levels were higher in G1 than in G2. Furthermore, in contrast to the positive relationship between hepatic functional reserve and degree of PEM, no relationship was found between severity of liver dysfunction and degree of insulin resistance in patients with cirrhosis [24]. These findings support the notion that LES with BCAA-EN does not always inhibit overt diabetes in patients with cirrhosis. However, further studies of the effects of LES treatment on glucose metabolism in patients with cirrhosis and impaired glucose tolerance must be conducted to validate the results of the current study.

This study had several limitations. First, the sample size was extremely small owing to the complexity of the FGM system. Second, the duration of BCAA-EN administration was extremely short. However, a nutrition study with larger sample size and longer treatment duration is challenging to perform on high-risk patients with cirrhosis. Moreover, concerns have been raised as one patient withdrew from the study due to the poor palatability of BCAA-EN. Recently, premixed flavored formulations have been developed to make the formulations more palatable. The ready-to-drink formulations will improve compliance to BCAA formulations at least to some extent.

In conclusion, the administration of LES with BCAA-EN does not inhibit overt diabetes in most patients with cirrhosis. The average blood glucose levels after LES treatment significantly increased in patients with obesity and severe insulin resistance whose nocturnal BG levels were <200 mg/dL. Careful attention must be paid to fluctuations in BG levels in patients with cirrhosis who present with obesity and severe insulin resistance. This study suggests that nocturnal BCAA-EN administration as a LES is generally safe and without side effects for most patients with cirrhosis; however, it is more suitable for those without obesity and severe insulin resistance. Nutritionists provide nutritional support to improve glucose metabolism during LES treatment for patients with cirrhosis and glucose intolerance. Moreover, larger studies must be conducted to evaluate the effects of LES with BCAA-EN on glucose metabolism in patients with cirrhosis.

## Figures and Tables

**Figure 1 nutrients-11-02140-f001:**
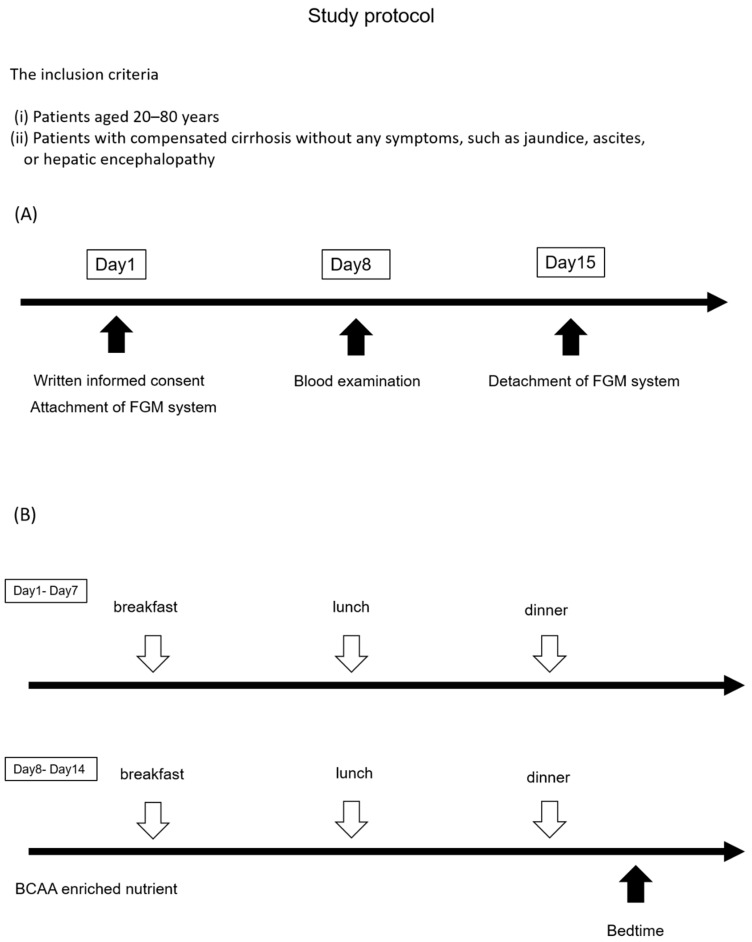
Study protocol: (**A**) Blood glucose (BG) levels were determined every 15 min for 14 days (day1–day14) with the Flash Glucose Monitoring (FGM) system. The FMG system was attached on day 1 and detached on day 15. The patients underwent blood examination on day 8. (**B**) BCAA-EN was administered at bedtime for 1 week (day 8–day 14) in all 13 patients with cirrhosis. BCAA, branched-chain amino acid; FMG, Flush Glucose Monitoring.

**Figure 2 nutrients-11-02140-f002:**
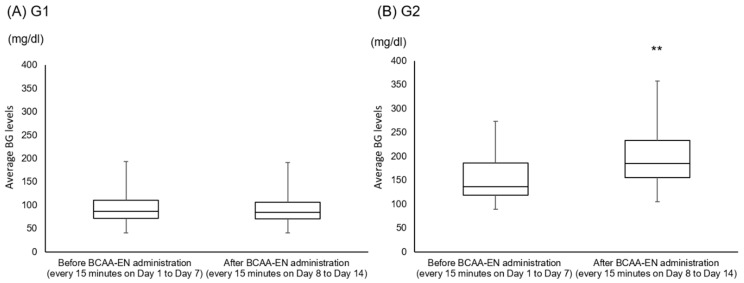
Average BG levels before (every 15 min in the nighttime between days 1 and 7) and after (every 15 min in the nighttime between days 8 and 14) BCAA-EN administration in patients with cirrhosis. The average BG levels did not significantly change after BCAA-EN administration in G1 (before 91.9 ± 29.0 mg/dL; after 89.0 ± 24.3 mg/dL). (**B**) However, the average BG levels significantly increased after BCAA-EN administration in G2 (before 153.6 ± 43.3 mg/dL; after 200.9 ± 59.7 mg/dL). G1, group 1: patients whose nocturnal BG levels were ≥200 mg/dL; G2, group 2: patients whose nocturnal BG levels were <200 mg/dL; BCAA-EN, branched-chain amino acid-enriched nutrient; BG, blood glucose. Asterisks indicate statistically significant differences between groups 1 and 2 (** *p* < 0.01).

**Table 1 nutrients-11-02140-t001:** Clinical and demographic characteristics of patients with cirrhosis.

Parameters	Total	G1: Nocturnal blood glucose <200mg/dL (*n* = 11)	G2: Nocturnal blood glucose ≥ 200mg/dL (*n* = 2)	P value
**Age (years old)** **^a^**	69.8 ± 12.3	71.2 ± 11.9	61.5 ± 11.5	0.016
**Gender (Male/Female)**	7/6	5/6	2/0	0.224
**BMI** **^a^**	23.7 ± 4.0	22.5 ± 2.8	30.1 ± 3.8	0.268
**Etiology HBV/HCV/Alcohol /others**	1/5/5/2	1/5/3/2	0/0/2/0	0.286
**Child-pugh score**	9/4	6.1 ± 1.2	5.5 ± 0.5	0.027
**Platelet (×10^4^/μL)** **^a^**	9.3 ± 4.6	9.8 ± 4.8	6.7 ± 2.1	0.178
**PT** **(%) ^a^**	72.6 ± 14.4	73.0 ± 15.5	70.5 ± 4.5	0.050
**CRP** **(mg/dL)** **^a^**	0.10 ± 0.07	0.09 ± 0.08	0.14 ± 0.03	0.899
**TP (g/dL)** **^a^**	7.3 ± 0.5	7.2 ± 0.5	7.7 ± 0.1	0.027
**Alb (g/dL)** **^a^**	3.9 ± 0.5	3.9 ± 0.5	4.1 ± 0.1	0.032
**BTR** **^a^**	5.76 ± 4.39	5.86 ± 4.69	5.21 ± 1.87	0.538
**GGT (IU/L)** **^a^**	54.3 ± 57.6	52.2 ± 62.4	66.0 ± 4.0	0.733
**T-Bil** **(IU/L) ^a^**	1.29 ± 0.61	1.32 ± 0.65	1.15 ± 0.15	0.233
**UA** **(mg/dL) ^a^**	5.97 ± 1.30	5.69 ± 1.23	7.50 ± 0.00	0.236
**NH3 (** **μg/dL)** **^a^**	53.9 ± 28.7	48.0 ± 23.4	05.9 ± 5.4	0.924
**FBG (mg/dL)** **^a^**	104.9 ± 23.7	98.6 ± 19.9	139.5 ± 8.5	0.370
**HbA1c** **(%) ^a^**	5.41 ± 0.81	5.2 ± 0.65	6.55 ± 0.65	0.169
**Glycated albumin** **(%)** **^a^**	16.2 ± 2.0	16.2 ± 2.1	16.6 ± 0.9	0.033
**F-IRI** **(μU/mL) ^a^**	22.2 ± 21.5	16.5 ± 11.0	53.9 ± 34.0	0.012
**HOMA-IR** **^a^**	6.15 ± 6.91	4.02 ± 2.59	17.85 ± 10.58	0.001
**QUICKI** **^a^**	0.31 ± 0.03	0.32 ± 0.03	0.27 ± 0.02	0.002
**LDL-cho/HDL-cho** **^a^**	1.34 ± 0.63	1.21 ± 0.35	2.01 ± 1.18	0.234

^a^ Mean ± standard error of mean. BMI: body mass index, PT: prothrombin time, CRP: C-reactive protein, TP: total protein, Alb: albumin, BTR: BCAA/Tyrosine ratio, GGT: γ-glutamyl transpeptidase, T-Bil: total bilirubin, UA: uric acid, NH3: ammonia, FBG: fasting blood glucose, HbA1c: Hemoglobin A1c, GA: glycated albumin, F-IRI: fasting immunoreactive insulin, HOMA IR: Homeostasis model assessment of insulin resistance, QUICKI: Quantitative insulin sensitivity check index, LDL-cho: low density lipoprotein cholesterol, HDL-cho: high density lipoprotein cholesterol.

**Table 2 nutrients-11-02140-t002:** Clinical profiles of patients with cirrhosis.

Case	Age (years)	SEX (F/M)	BMI (kg/m^2^)	CP score	FBG (mg/dL)	HbA1c (%)	GA (%)	F-IRI (μU/mL)	HOMA-IR	QUICKI
**1**	88	F	23.4	8	87	4.3	19.1	8.7	1.87	0.35
**2**	69	M	27.5	7	102	4.5	17.5	18.8	4.73	0.30
**3**	67	M	22.8	7	94	4.7	15.4	14.4	3.34	0.32
**4**	89	M	18.2	5	78	5.4	15.1	9	1.73	0.35
**5**	64	F	23.9	6	94	4.4	14.9	21	4.87	0.30
**6**	77	F	20.4	5	81	4.9	12.6	47.4	9.48	0.28
**7**	54	F	18.7	5	90	5.6	14.1	7.1	1.58	0.36
**8**	76	F	26.4	5	85	5.4	14.4	11.4	2.39	0.33
**9**	49	M	21.4	8	108	5.8	19.1	12	3.20	0.32
**10**	75	F	21.1	5	114	6.3	16.9	8.7	2.45	0.33
**11**	76	M	24.1	6	152	5.9	18.8	22.9	8.59	0.28
**12**	73	M	26.3	5	148	7.2	17.5	19.9	7.27	0.29
**13**	50	M	33.8	6	131	5.9	15.7	87.9	28.43	0.25

BMI: body mass index, CP: Child Pugh score, FPG: fasting blood glucose, HbA1c: Hemoglobin A1c, GA: glycated albumin, F-IRI: fasting immunoreactive insulin, HOMA IR: Homeostasis model assessment of insulin resistance, QUICKI: Quantitative insulin sensitivity check index, N.A: not applicable.

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
