# Peer review of "Late-Evening Snack with Branched-Chain Amino Acid-Enriched Nutrients Does Not Always Inhibit Overt Diabetes in Patients with Cirrhosis: A Pilot Study"

_nutrients, 2019, doi:10.3390/nu11092140_

Round 1

Reviewer 1 Report

Using a figure to help explain the patient inclusion criteria will be better. Please provide higher quality of the figure, the resolution is too low. Please provide the project identification code in the ethical statement on line 74. The author indicated that the BG levels were determined every 15 minutes (line 67), however, in figure 2 there were only two dots that indicate the BG levels before and after the BCAA-EN administration. The author should provide more specific data (ex: the time point, the continued BG levels) to clearly clarify the result. 

Author Response

August 23, 2019

Nutrients

Manuscript ID: nutrients-586446

Title: Late-evening snack with branched-chain amino acid-enriched nutrient does not always inhibit overt diabetes in patients with cirrhosis: a pilot study

Authors: Keisuke Nakanishi, Tadashi Namisaki, Tsuyoshi Mashitani, Kosuke Kaji, Kuniaki Ozaki, Soichiro Saikawa, Shinya Sato, Takashi Inoue, Yasuhiko Sawada, Kou Kitagawa, Hiroaki Takaya, Naotaka Shimozato, Hideto Kawaratani, Kei Moriya, Takemi Akahane, Akira Mitoro, and Hitoshi Yoshiji

We are very grateful to you and the reviewers for the helpful comments and suggestions regarding our manuscript. We have thoroughly addressed all concerns and issues raised and have accordingly revised our manuscript. All changes in the revised manuscript are highlighted in yellow. We believe that the manuscript has been greatly improved and hope it is ready for publication in Nutrients. Once again, we acknowledge your comments that have been extremely valuable in improving the quality of our manuscript. We have provided point-by-point responses to the reviewers’ comments below.

Reviewer: 1 Comments and Suggestions for Authors

Using a figure to help explain the patient inclusion criteria will be better. Please provide higher quality of the figure, the resolution is too low. Please provide the project identification code in the ethical statement on line 74. The author indicated that the BG levels were determined every 15 minutes (line 67), however, in figure 2 there were only two dots that indicate the BG levels before and after the BCAA-EN administration. The author should provide more specific data (ex: the time point, the continued BG levels) to clearly clarify the result.

Author response: We greatly appreciate your constructive comments. We apologize for having shown wrong study period. The new figure 1 shows an appropriate period. We have revised our manuscript accordingly. Blood glucose (BG) levels were determined every 15 min for 14 days (day1–14) with the Flash Glucose Monitoring (FGM) system (Abbott Diabetes Care, Alameda, CA) (New reference No.10 Dover, J Diabetes Sci Technol 2017) (Fig. 1A). BCAA-EN was administered at bedtime for 1 week (day 8–day 14) in 13 (8.1%) of the 160 patients (Fig. 1B). We have included a description of these findings on page 2, lines 71–74 and page 3, lines 85–88.

Reference

Dover A.R.; Stimson R.H.; Zammitt N.N.; Gibb F.W. Flash Glucose Monitoring Improves Outcomes in a Type 1 Diabetes Clinic. J Diabetes Sci Technol 2017;11:442–443.

Reviewer 2 Report

Nutrients BCAA

Although a potential interesting article, there are several issues the authors must address.

[*]  Abstract. It is inappropriate to simply list p-values. You must list real data and appropriate error terms along with your prioritized outcomes (see next comment).

[*] The Author Instructions state:

“CONSORT Statement

Nutrients requires a completed CONSORT 2010 checklist and flow diagram as a condition of submission when reporting the results of a randomized trial. Templates for these can be found here or on the CONSORT website (http://www.consort-statement.org) which also describes several CONSORT checklist extensions for different designs and types of data beyond two group parallel trials. At minimum, your article should report the content addressed by each item of the checklist. Meeting these basic reporting requirements will greatly improve the value of your trial report and may enhance its chances for eventual publication.”

-Outcomes 6a Completely defined pre-specified primary and secondary outcome measures, including how and when they were assessed.

These are not defined.

-Outcomes and estimation 17a For each primary and secondary outcome, results for each group, and the estimated effect size and its precision (such as 95% confidence interval).

These are not presented.

[*] Line 75. Please present the University approval number unless Nutrients have asked you to withhold it for the review

[*]  Line 81. More is needed on the statistics. Please explain why you used a Mann-Whitnney U test instead of a GLM.  A GLM is robust to issues surrounding normality and enable you to adjust for sex and age and other factors. 

Consideration should be given for:

Age

Sex

Years presenting with cirrhosis. In essence, a chronic vs. acute diagnosis could make a difference in you findings.

You may not use them, but they should be analyzed, reported and accounted for if necessary

[*]  Line 84. Please state how our data are presented. Data are presented as mean (SD) or mean/percent change (95%). I don’t see this presented anywhere. 

Data should be presented as:

Mean (SD)

Mean or Percent change (95% CI)

SEM should not be used in the text or in the figures in an intervention trial.

Effect sizes should be presented throughout.

Author Response

August 23, 2019

Nutrients

Manuscript ID: nutrients-586446

Title: Late-evening snack with branched-chain amino acid-enriched nutrient does not always inhibit overt diabetes in patients with cirrhosis: a pilot study

Authors: Keisuke Nakanishi, Tadashi Namisaki, Tsuyoshi Mashitani, Kosuke Kaji, Kuniaki Ozaki, Soichiro Saikawa, Shinya Sato, Takashi Inoue, Yasuhiko Sawada, Kou Kitagawa, Hiroaki Takaya, Naotaka Shimozato, Hideto Kawaratani, Kei Moriya, Takemi Akahane, Akira Mitoro, and Hitoshi Yoshiji

We are very grateful to you and the reviewers for the helpful comments and suggestions regarding our manuscript. We have thoroughly addressed all concerns and issues raised and have accordingly revised our manuscript. All changes in the revised manuscript are highlighted in yellow. We believe that the manuscript has been greatly improved and hope it is ready for publication in Nutrients. Once again, we acknowledge your comments that have been extremely valuable in improving the quality of our manuscript. We have provided point-by-point responses to the reviewers’ comments below.

Reviewer: 2 Comments and Suggestions for Authors

Nutrients BCAA

Although a potential interesting article, there are several issues the authors must address.

[*]  Abstract. It is inappropriate to simply list p-values. You must list real data and appropriate error terms along with your prioritized outcomes (see next comment).

[*] The Author Instructions state:

Author response: Thank you for your valuable comments. According to your comments, we have revised this and included a description of these findings on page 1, lines 21–29.

“CONSORT Statement

Nutrients requires a completed CONSORT 2010 checklist and flow diagram as a condition of submission when reporting the results of a randomized trial. Templates for these can be found here or on the CONSORT website (http://www.consort-statement.org) which also describes several CONSORT checklist extensions for different designs and types of data beyond two group parallel trials. At minimum, your article should report the content addressed by each item of the checklist. Meeting these basic reporting requirements will greatly improve the value of your trial report and may enhance its chances for eventual publication.”

-Outcomes 6a Completely defined pre-specified primary and secondary outcome measures, including how and when they were assessed.

These are not defined.

 Author response: Thank you for your valuable comments. Blood glucose (BG) levels were determined every 15 min for 14 days (day1–day14) with the Flash Glucose Monitoring (FGM) system (Abbott Diabetes Care, Alameda, CA)(New reference No.10 Dover, J Diabetes Sci Technol 2017)(Fig. 1A). Branched-chain amino acid-enriched nutrient (BCAA-EN) was administered at bedtime for 1 week (day 8-day 14) in 13 (8.1%) of the 160 patients (Fig. 1B). The patients were divided into two groups based on BG levels: group 1 (G1, n = 11): casual BG levels < 200 mg/dL and group 2 (G2, n = 2): casual BG levels ≥ 200 mg/dL. The changes in BG levels before and after BCAA-EN administration were compared between G1 and G2 groups. Primary outcome is suppression of elevation of BG levels after BCAA-EN administration. We have included a description of these findings on page 2, lines 71–79.

-Outcomes and estimation 17a For each primary and secondary outcome, results for each group, and the estimated effect size and its precision (such as 95% confidence interval).

These are not presented.

 Author response: Thank you for your valuable comments. Primary outcome is suppression of elevation of BG levels after branched-chain amino acid enteral nutrient (BCAA-EN) administration. The changes in BG levels before and after BCAA-EN administration were compared between group 1 (G1: patients whose casual BG levels were<200 mg/dL) and group 2 (patient whose casual BG levels were ≥200 mg/dL). The average BG levels did not significantly change after BCAA-EN administration in G1 (before 91.9 ± 29.0 mg/dl and after 89.0 ± 24.3 mg/dl) (Fig.2A). However, the average BG levels significantly increased after BCAA-EN administration in G2 (before 153.6 ± 43.3 mg/dl and after 200.9 ± 59.7 mg/dl) (p < 0.01) (Fig.2B). The estimated effect size is the differences in blood glucose levels before and after BCAA-EN administration. This study suggests that nocturnal BCAA-EN administration as LES is generally safe and without side effects for most patients with cirrhosis; however, it is more suitable for those without obesity and severe insulin resistance. We revised the figure 2 for the precision. We have included a description of these findings on page 2, lines 76–79, page 5, lines 135–140 and page 7, lines 182–184.

[*] Line 75. Please present the University approval number unless Nutrients have asked you to withhold it for the review

Author response: Thank you for your valuable comments. We have accordingly included a description about this on page 2, line 83.

[*] Line 81. More is needed on the statistics. Please explain why you used a Mann-Whitnney U test instead of a GLM.  A GLM is robust to issues surrounding normality and enable you to adjust for sex and age and other factors. 

Consideration should be given for:

Age

Sex

Years presenting with cirrhosis. In essence, a chronic vs. acute diagnosis could make a difference in you findings.

You may not use them, but they should be analyzed, reported and accounted for if necessary

Author response: Thank you for your valuable comments. The sample size is too small to analyze the data using the generalized linear model. Mann–Whitnney U test was used in this study for comparing the baseline characteristics between group 1 (patients whose casual BG levels were <200 mg/dL) and group 2 (patients whose casual BG levels were ≥200 mg/dL). Wilcoxon signed-rank test was used to compare the changes in BG levels before and after BCAA-EN administration between the two groups. We have included a description of these findings on page 3, lines 92–96.

[*]  Line 84. Please state how our data are presented. Data are presented as mean (SD) or mean/percent change (95%). I don’t see this presented anywhere. 

Data should be presented as:

Mean (SD)

Mean or Percent change (95% CI)

SEM should not be used in the text or in the figures in an intervention trial.

Effect sizes should be presented throughout.

Author response: Thank you for your valuable comments. Data are presented as mean ± standard deviation. The estimated effect size is the differences in blood glucose levels before and after branched-chain amino acid-enriched nutrient administration. We have included a description of these findings on page 3, lines 91 and page 2, lines 78–79.

Reference

Dover A.R.; Stimson R.H.; Zammitt N.N.; Gibb F.W. Flash Glucose Monitoring Improves Outcomes in a Type 1 Diabetes Clinic. J Diabetes Sci Technol 2017;11:442–443.

Round 2

Reviewer 1 Report

The author revised most question. However, In figure 2, the author described the changes in BG levels before and after branched-chain amino acid enteral nutrient administration. However, it is important to provide the exact timing of sample collection. For example, the definition of before BCAA-EN administration means the BG levels analyzed “10 minutes” before BCAA-EN administration. On the other hand, the after BCAA-EN administration means the BG levels analyzed “10 minutes” after BCAA-EN administration. The author should provide the exact timing when they measured the blood glucose levels. 

Author Response

August 30, 2019

Nutrients

Manuscript ID: nutrients-586446

Originally titled: Late-evening snack with branched-chain amino acid-enriched nutrient does not always inhibit overt diabetes in patients with cirrhosis: a pilot study

Authors: Keisuke Nakanishi, Tadashi Namisaki, Tsuyoshi Mashitani, Kosuke Kaji, Kuniaki Ozaki, Soichiro Saikawa, Shinya Sato, Takashi Inoue, Yasuhiko Sawada, Kou Kitagawa, Hiroaki Takaya, Naotaka Shimozato, Hideto Kawaratani, Kei Moriya, Takemi Akahane, Akira Mitoro, and Hitoshi Yoshiji

We are very grateful to you and the reviewers for the helpful comments and suggestions regarding our manuscript. We have thoroughly addressed all concerns and issues raised and have accordingly revised our manuscript. All changes in the revised manuscript are highlighted in yellow. We believe that the manuscript has been greatly improved and hope it is ready for publication in Nutrients. Once again, we acknowledge your comments that have been extremely valuable in improving the quality of our manuscript. We have provided point-by-point responses to the reviewers’ comments below.

Reviewer: 1 Comments and Suggestions for Authors

The author revised most question. However, In figure 2, the author described the changes in BG levels before and after branched-chain amino acid enteral nutrient administration. However, it is important to provide the exact timing of sample collection. For example, the definition of before BCAA-EN administration means the BG levels analyzed “10 minutes” before BCAA-EN administration. On the other hand, the after BCAA-EN administration means the BG levels analyzed “10 minutes” after BCAA-EN administration. The author should provide the exact timing when they measured the blood glucose levels.

Author response: Thank you for your valuable comments. According to your comments, we have revised figure 2 and our manuscript, accordingly. Changes in BG levels before (every 15 minutes in the daytime and nighttime between days 1 and 7) and after (every 15 minutes in the daytime and nighttime between days 8 and 14) BCAA-EN administration were compared between G1 and G2. We have included a description of these findings on page 2, lines 76–78 and page 6, lines 118–119.

Reviewer 2 Report

n/a

Author Response

N/A